# pFedKT: Personalized Federated Learning via Knowledge Transfer

## Abstract

Federated learning (FL) has been widely studied as a new paradigm to achieve multi-party collaborative modelling on decentralized data with privacy protection. Unfortunately, traditional horizontal FL suffers from Non-IID data distribution, where clients' private models after FL are even inferior to models trained standalone. To tackle this challenge, most existing approaches focus on personalized federated learning (PFL) to improve personalized private models but present limited accuracy improvements. To this end, *we design pFedKT, a novel personalized federated learning framework with private and global knowledge transfer, towards boosting the performances of personalized private models on Non-IID data*. It involves two types of knowledge transfer: a) transferring *historical private knowledge* to new private models by local hypernetworks; b) transferring *the global model's knowledge* to private models through contrastive learning. After absorbing the historical private knowledge and the latest global knowledge, the personalization and generalization of private models are both enhanced. Besides, we derive pFedKT's generalization and prove its convergence theoretically. Extensive experiments verify that pFedKT presents $1.38\% - 1.62\%$ accuracy improvements of private models compared with the state-of-the-art baseline.

## 1 Introduction

With frequent privacy leakage, directly collecting data and modelling it would violate privacy protection regulations such as GDPR (Kairouz & et al., 2021). To implement collaborative modelling while protecting data privacy, horizontal federated learning (FL) came into being (McMahan & et al, 2017). As shown in Fig. 1 (a), FL consists of a central server and multiple clients. In each communication round, the server broadcasts the global model (abbr. GM) to selected clients; then clients train it locally on their local datasets and upload trained private models (abbr. PMs) to the server; finally, the server aggregates received private models to update the global model. The whole procedure is repeated until the global model converges. In short, FL fulfils collaborative modelling by allowing clients to only communicate model updates with the server, while data is always stored locally.

However, FL still faces several challenges such as communication efficiency, robustness to attacks, and *model accuracy* which we focus on in this work. The motivation for clients to participate in FL is to improve their local models' quality. However, the decentralized data held by clients are often not independent and identically distributed (Non-IID) (Kairouz & et al., 2021), and the global model aggregated through a typical FL algorithm FedAvg (McMahan & et al, 2017) based on Non-IID data may perform worse than clients' solely trained models. Zhao & et al (2018) have verified this fact experimentally and argued that the global model aggregated by skewed local models trained on Non-IID data deviates from the optima (model trained on *all* local data). To alleviate the accuracy degradation caused by Non-IID data, personalized FL (PFL) methods (Shamsian & et al, 2021) have been widely studied to improve clients' personalized model quality.

Existing researches implement PFL by fine-tuning Mansour & et al (2020); Wang & et al (2019), model mixup Arivazhagan & et al (2019); Collins & et al (2021) and etc. But they suffer from limited improvements in the accuracy of private models.

To further improve personalized private models on Non-IID data, we propose a novel personalized FL framework named *pFedKT* with two types of transferred knowledge: 1) *private knowledge*: we deploy a local hypernetwork for each client to transfer historical PMs' knowledge to new PMs; 2) *global knowledge*: we exploit contrastive learning to enable PMs to absorb the GM's knowledge. We analyzed pFedKT's generalization and proved its convergence theoretically. We also conducted extensive experiments to verify that pFedKT fulfils the state-of-the-art PM's accuracy.

**Contributions.** Our main contributions are summarized as follows: a) We devised two types of knowledge transfer to simultaneously enhance the generalization and personalization of private models. b) We analyzed pFedKT's generalization and convergence in theory. c) Extensive experiments verified the superiority of pFedKT on the accuracy of personalized private models.

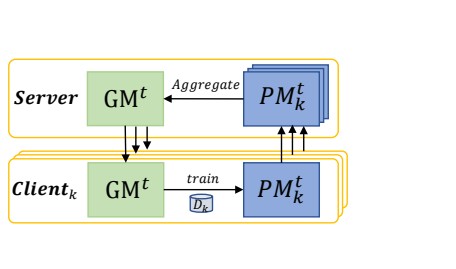
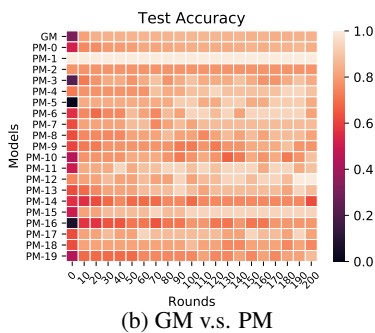

(a) FedAvg        (b) GM v.s. PM

Figure 1: (a): Workflow of FedAvg in the $t$-th round. (b): The test accuracy of the GM and 20 PMs are recorded per 10 rounds. Since the server has no data, we evaluate the test accuracy of the GM on clients' test datasets and calculate mean test accuracy as the GM's accuracy. We evaluate the test accuracy of a local model $PM$ *after local training* on its local test data as the PM's accuracy.

## 2 RELATED WORK

Recent personalized federated learning (PFL) approaches include: **a) Fine-tuning**, in FL's last round, clients fine-tune the received GM on local data to get PMs (Wang & et al, 2019; Mansour & et al, 2020). **b) Federated meta-learning**, some methods apply meta-learning in FL, such as MAML-based distributed variants (Li & et al, 2017; Fallah & et al, 2020b;a). **c) Federated multi-task learning**, it treats each client as a learning task, e.g., MOCHA (Smith & et al, 2017), FedU (Dinh & et al, 2021). **d) Model mixup**, the PM's parameters are split into two parts, only one part is shared through the server and another is trained locally, as in FedPer (Arivazhagan & et al, 2019), FedFu (Yao et al., 2019), FLDA (Peterson et al., 2019), LG-FEDAVG (Liang & et al, 2020), MAPPER (Mansour et al., 2020), FedRep (Collins & et al, 2021), pFedGP (Achituve & et al, 2021), (Sun & et al, 2021). **e) Aggregation Delay**, RADFed Xue et al. (2021) proposed redistribution rounds that delay aggregation to alleviate the negative impacts on model performance due to Non-IID data. **f) Federated clustering**, the server clusters PMs with similar parameter distributions and performs aggregation within clusters, e.g., HYPCLUSTER (Mansour & et al, 2020), ClusterFL (Ouyang & et al, 2021), CFL (Agrawal & et al, 2021). **g) Local aggregation**, instead of aggregation within the server's clustered groups, FedFOMO (Zhang & et al, 2021) makes each client pull other clients' PMs and selects more beneficial ones for local aggregation to update its own PMs. **h) Knowledge distillation-based**, FedPHP (Li & et al., 2021) linearly accumulates historical PMs and new trained PMs to teach the received GM through knowledge distillation in each round of FL. FML (Shen et al., 2020) makes each client's PM interact with the GM through mutual learning. KT-pFL (Zhang et al., 2021) allocates a public dataset to each client, and only logits computed on the public dataset are shared through the server. **i) Contrastive learning-based**, MOON (Li et al., 2021) utilizes contrastive learning to make PMs close to the GM, towards obtaining a better GM. **j) Hypernetwork-based**, pFedHN Shamsian & et al (2021) deploys a *global* hypernetwork on the server to learn PMs' parameter distributions and generate personalized parameters for PMs. The latest work Fed-RoD (Chen & Chao, 2022) trains private personalized *headers* with parameters generated by local hypernetworks. It improves both the GM and PMs, but extra communication cost incurs by communicating hypernetworks.

## 3 PRELIMINARIES AND MOTIVATION

### 3.1 UTILITY OF PRIVATE MODELS

As the workflow of FedAvg shown in Fig.1 (a), we abbreviate the private model as **PM** and the global model as **GM**, and the detailed definition of FL is introduced in Appendix A. It's worth noting that: in FedAvg, 1) clients no longer store PMs after uploading them to the server; 2) in the next round, clients regard *the received GM* as PM and then train PM on local datasets, i.e., the trained PMs only play as "temporary models" for aggregation and their utilities are not sufficiently developed.

To explore the utilities of PMs, we train a CNN model on a natural Non-IID FEMINIST dataset in an FL system with 20 clients. From Fig.1 (b), we observe that there are always some PMs performing better than GM in each round (some PMs show lighter pixels than GM), so *we can further develop PMs' self-utility during FL to boost the accuracy of personalized private models*.

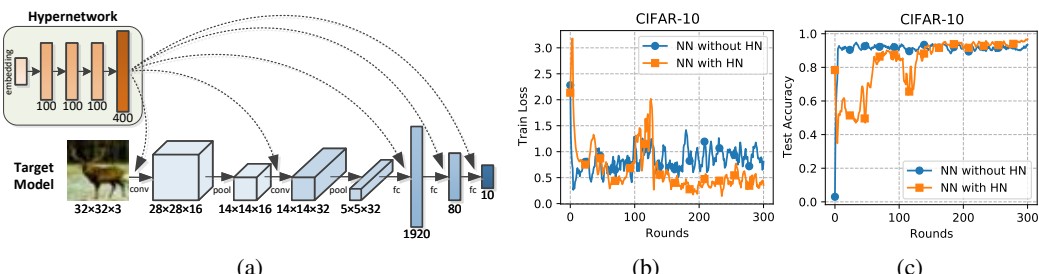

(a)                                       (b)                                       (c)

Figure 2: (a): A hypernetwork generates parameters for a target model. (b): the test accuracy of two target models (NN with/out HN) vary as communication rounds on CIFAR-10 dataset.

## 3.2 MOTIVATION

### 3.2.1 PRIVATE KNOWLEDGE ENHANCEMENT

To fully utilize PMs' self-utility, FedPHP *linearly accumulates* historical PMs and new trained PMs to teach the received GM through knowledge distillation in each round of FL. Since historical private models with obsolete parameters compromise convergence, linearly stacking them with manually selected weights may suffer from degraded accuracy, which has been verified in our subsequent experiments (Sec. 6.2). A hypernetwork (abbr. HN) (Ha & et al, 2017) is generally a small model that generates parameters for large target models, belonging to the category of *generative networks* in unsupervised learning. pFedHN and Fed-RoD exploit hypernetworks to personalize PMs, but they have a few weaknesses. In pFedHN, the footprint of the server's hypernetwork is designed to be larger than that of PMs, especially, the hypernetwork's output layer's parameter capacity is equal to that of a complete PM, which burdens the storage cost and computation overhead of the server. Fed-RoD requires to communicate local hypernetworks (which generate parameters for personalized *headers*) between clients and the server, introducing extra communication cost which is the main bottleneck of FL (Kairouz & et al., 2021).

To enhance personalization while avoiding the above problems, our pFedKT attempts to allocate a *local hypernetwork* for each client to learn its PM's *historical private knowledge* (parameter distributions) and then generate parameters for its *complete* private model. In other words, we use the *local* hypernetwork to accumulate historical PM's knowledge to develop PMs' self-utility. The clients' local hypernetworks used in our pFedKT are simple fully connected networks with lower parameter capacity than PMs, so it has lower computation complexity than the server's large hypernetwork in pFedHN. Besides, the server and clients still communicate PMs in pFedKT, which has a communication cost the same as FedAvg and lower than Fed-RoD.

To verify the feasibility of the above insight, we conduct preliminary experiments on a *single* client. Specifically, we train a randomly initialized hypernetwork (abbr. HN) and a target model (abbr. NN, i.e., PM) with parameters generated by the hypernetwork in an end-to-end form on the CIFAR-10 dataset. How to use hypernetworks to generate parameters for target models and how to update hypernetwork are detailed in Appendix B and C. Fig. 2 displays the structures of the target model and hypernetwork, as well as the experimental results. We observe that the final test accuracies of the solely trained target model (NN without HN) and the target model trained with the hypernetwork (NN with HN) are $91.84\%$ and $93.84\%$, respectively, i.e., the latter performs better, indicating that regarding HN as a meta-model to continually learn PM's historical knowledge boosts PM's accuracy. Therefore, *we can safely utilize the local hypernetwork to accumulate PM's private knowledge*.

### 3.2.2 GLOBAL KNOWLEDGE ENHANCEMENT

To train a better GM on Non-IID data, MOON Li et al. (2021) utilizes contrastive learning to keep the current round's PM (anchor) close to the received GM (positive) and away from the last round's PM (negative). Since it initializes the current round's PM (anchor) with GM's parameters, the PM's personalization is impaired. Nevertheless, *it still Inspires us to transfer global knowledge to PMs*.

Unlike MOON, our pFedKT regards *the PM generated by a hypernetwork which has absorbed historical private knowledge* as the anchor in *contrastive learning*. And it also keeps the generated PM close to GM for acquiring global knowledge. This discrepancy facilitates the effective fusion of the *latest global* knowledge and *historical private* knowledge, promoting PM's personalization and generalization in available classes of local datasets.

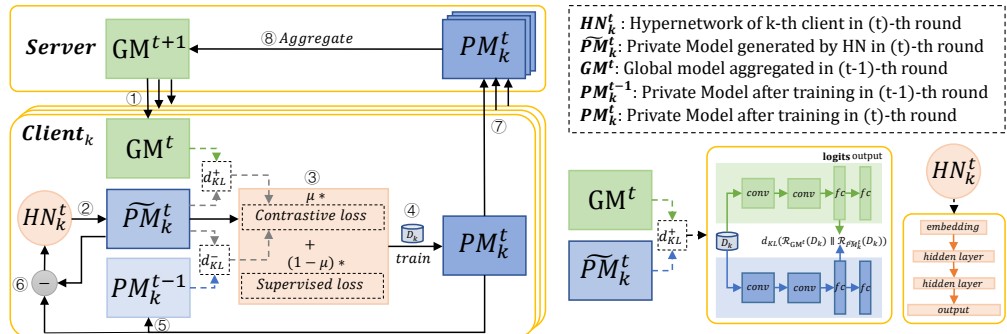

Figure 3: Left: workflow of pFedKT. Right: KL divergence between two models is calculated on logits $\mathcal{R}_{\omega\cdot}(D_k)$; and the local hypernetwork used in pFedKT is a simple fully connected (FC) network with much smaller footprint than that of PM.

## 4 METHODOLOGY

In this section, we first outline pFedKT's workflow, and then detail the two types of knowledge transfer: a) private knowledge transfer by local hypernetworks and b) global knowledge transfer by contrastive learning. Finally, we analyze pFedKT's computational budgets and storage costs.

### 4.1 OVERVIEW

**Principle**. Building on the above motivations, we devise a novel personalized FL framework named pFedKT, which involves two types of knowledge transfer: 1) **Transferring knowledge from old PM to new PM**. We configure a local hypernetwork (with much smaller footprint than that of PM) for each client to learn the old PM's knowledge and transfer it to the new PM. 2) **Transferring knowledge from GM to PM**. We exploit contrastive learning to keep the new PM (have carried old private knowledge) close to GM. In this way, during each round of local training, *the private knowledge from the old PM*, *the global knowledge from the latest GM*, and absolutely *the knowledge from local data* are simultaneously incorporated into the trained new PM, which facilitates personalization and generalization in available classes.

**Workflow**. Specifically, as displayed in Fig. 3, the complete workflow of pFedKT framework includes the following steps: in the $t$-th round, a) the server first broadcasts the global model $GM^t$ to selected clients. 2) The $k$-th client uses its local hypernetwork $HN_k^t$ (with the embedding of our pre-divided chunk id as input) to generate parameters for the target private model $PM_k^{t_0}$ (to be trained) in a stacked manner, where we detailed the generation procedure in Appendix B. 3) Then, we regard the generated private model $PM_k^{t_0}$ as the anchor, the received global model $GM^t$ as a positive item, the last round's trained private model $PM_k^{t-1}$ as a negative item. The $k$-th client computes the distances $d_{KL}^+$ and $d_{KL}^-$ measured by KL (Kullback-Leibler) divergence Kullback & Leibler (1951) from $PM_k^{t_0}$ to $GM^t$ and $PM_k^{t-1}$, respectively. After that, the $k$-th client computes the *contrastive loss* with the two distances. 4) The $k$-th client trains $PM_k^{t_0}$ with *contrastive loss* and *supervised loss* from labels on local dataset $D_k$. After training, the trained private model is marked as $PM_k^t$. 5) The $k$-th client updates the old private model $PM_k^{t-1}$ with the latest trained private model $PM_k^t$. 6) The $k$-th client updates the hypernetwork $HN_k^t$ to $HN_k^{t+1}$ with the parameter variations between generated $PM_k^{t_0}$ and trained $PM_k^t$, where we detailed the hypernetwork's updating procedure in Appendix C. 7) The $k$-th client uploads the trained private model $PM_k^t$ to the server. 8) The server aggregates the received private models $[PM_k^t, \cdots]$ through the weighted aggregation rule of FedAvg, and updates the global model to be $GM^{t+1}$. The above steps are executed iteratively until PMs converge. In the end, we acquire *personalized PMs*. A detailed algorithm description of pFedKT is given in Appendix D, Alg. 1.

### 4.2 PRIVATE KNOWLEDGE TRANSFER VIA LOCAL HYPERNETWORKS

Motivated by the availability of hypernetworks validated in Sec. 3.2.1, we view hypernetworks as "information carriers" to continuously transfer *old private knowledge* of previous PMs to new PMs generated by hypernetworks. In particular, we deploy a local hypernetwork for each client. Utilizing hypernetworks to achieve private knowledge transfer involves two directions: a) *knowledge release (forward)* and b) *knowledge absorption (backward)*.

**Knowledge release (forward).** As step ② in Fig. 3, we first use the hypernetwork $\varphi_k^t$ to generate parameters for the new private model $\widetilde{\theta}_k^t$, in which hypernetworks *release* old knowledge of the last round's trained private models $\omega_k^{t-1}$ to the new generated model $\widetilde{\theta}_k^t$.

Then, the generated private model $\widetilde{\theta}_k^t$ will be trained on local dataset $D_k$, and the trained private model is denoted as $\omega_k^t$ which will be uploaded to the server for aggregation.

**Knowledge absorption (backward).** Instead of abandoning trained private models after uploading them to the server as in FedAvg, we use local hypernetworks to *absorb* the knowledge (parameter distribution) of trained private models. This procedure is implemented by updating hypernetworks' parameters with parameter variations between generated new PM and trained new PM, as step ⑥ in Fig. 3. Specifically, according to the rule of updating hypernetworks in Eq. (10) of Appendix C, we utilize the difference between the generated new PM $\widetilde{\theta}_k^t$ and trained new PM $\omega_k^t$ to update the local hypernetwork $\varphi_k^t$, i.e.,

$$\varphi_k^{t+1} \leftarrow \varphi_k^t - \eta_{HN}(\nabla_{\varphi_k^t}\theta_k^{t_0})^T\Delta(\theta_k^{t_0}, \omega_k^t). \tag{1}$$

where $\eta_{HN}$ is the hypernetwork's learning rate. Then the updated hypernetwork $\varphi_k^{t+1}$ absorbs the knowledge of the latest trained PM $\omega_k^t$.

Since the two-way knowledge transfer is executed in each round, hypernetworks continuously learn *historical private knowledge* and transfer it to the new generated PMs during the whole FL, which promotes the personalization of PMs.

### 4.3 GLOBAL KNOWLEDGE TRANSFER VIA CONTRASTIVE LEARNING

Once the new PM $\widetilde{\theta}_k^t$ is generated by the local hypernetwork, *old private knowledge* has been transferred into the new model. To make $\widetilde{\theta}_k^t$ further obtain *the latest GM's knowledge*, we exploit *contrastive learning* to bridge GM $\omega^t$ and PM $\widetilde{\theta}_k^t$. Specifically, we view $\widetilde{\theta}_k^t$ as the *anchor*, and keep it close to GM $\omega^t$ (*positive*) since we hope $\widetilde{\theta}_k^t$ to learn knowledge from other clients via GM; while keeping $\widetilde{\theta}_k^t$ away from the last round's PM $\omega_k^{t-1}$ (*negative*), so as to avoid being trapped in a local optimum and slowing down convergence due to excessively skewing previous PM's obsolete parameters. We use *triplet loss* (Schroff & et al., 2015) in typical contrastive learning as pFedKT's contrastive loss $\ell_{con}$. Then, we calculate contrastive loss by:

$$\mathcal{L}_{\omega^t} \leftarrow \mathcal{R}_{\omega^t}(D_k), \mathcal{L}_{\omega_k^t} \leftarrow \mathcal{R}_{\omega_k^t}(D_k), \mathcal{L}_{\omega_k^{t-1}} \leftarrow \mathcal{R}_{\omega_k^{t-1}}(D_k);$$
$$d_{KL}^+ = d_{KL}(\mathcal{L}_{\omega^t}\|\mathcal{L}_{\omega_k^t}), d_{KL}^- = d_{KL}(\mathcal{L}_{\omega_k^{t-1}}\|\mathcal{L}_{\omega_k^t}); \tag{2}$$
$$\ell_{con} = \max\{d_{KL}^+ - d_{KL}^- + \alpha, 0\}.$$

The distance $d_{KL}^+$ of $\widetilde{\theta}_k^t$ (anchor) and $\omega^t$ (positive), and the distance $d_{KL}^-$ of $\widetilde{\theta}_k^t$ (anchor) and $\omega_k^{t-1}$ (negative) are measured by the KL divergence of their logits $\mathcal{L}(\omega\cdot)$ (i.e., extracted representation $\mathcal{R}_{\omega\cdot}(D_k)$). $\alpha \geq 0$, is the maximum margin between the anchor-positive distance and anchor-negative distance. If $\alpha = 0$, the generated PM $\widetilde{\theta}_k^t$ is as far from GM $\omega^t$ as from the last round's PM $\omega_k^{t-1}$, i.e., "neutral" status. If $\alpha > 0$, then $d_{KL}^+ + \alpha \leq d_{KL}^-$, i.e., generated PM $\widetilde{\theta}_k^t$ is close to the GM $\omega^t$ and away the last round's PM $\omega_k^{t-1}$, and vice versa.

After computing contrastive loss $\ell_{con}$, we further calculate the supervised loss $\ell_{sup}$ (e.g., Cross Entropy loss) with the training model's predictions and labels. Finally, we linearly weight the two types of loss to build the complete loss function $f$, i.e.,

$$f = \mu * \ell_{con} + (1 - \mu) * \ell_{sup}, \tag{3}$$

where $\mu \in (0, 1)$ is the weight of contrastive loss. Next, we train the generated PM $\widetilde{\theta}_k^t$ on local data $D_k$ through gradient descent with the complete loss function $f$, and obtain trained PM $\omega_k^t$, i.e.,

$$\omega_k^t \leftarrow \theta_k^{t_0} - \eta_{NN}\nabla f(\theta_k^{t_0}; D_k), \tag{4}$$

where $\eta_{NN}$ is the private model's learning rate.

With contrastive learning, the trained PM $\theta_k^t$ absorbs the GM's global knowledge (i.e., private knowledge from other clients), enhancing the *generalization* of PMs in available classes of local data.

### 4.4 COMPUTATIONAL BUDGET AND STORAGE COST

In this section, we analyze the computational complexity and storage overhead of pFedKT compared with state-of-the-art pFedHN. Limited to pages, the detailed analysis and comparisons with more baselines are presented in Appendix G. After careful comparisons, we summarized as follows:

**Computational Complexity**. pFedKT consumes a comparable computational cost to pFedHN. In cross-silo FL scenarios, multiple enterprises with sufficient computational power can tolerate executing pFedKT. Besides, pFedKT inherently offloads the *serial* learning tasks of the hypernetwork on the *server* in pFedHN to *parallel clients' sub-tasks*, which reduces computation delay and tackles the *blocking issue* that possibly occurred on the server in pFedHN.

**Storage Overhead**. Since the local hypernetwork in pFedKT has a smaller footprint than the hypernetwork deployed on the server in pFedHN. Hence, pFedKT shows a lower storage cost than pFedHN from the perspective of one device (client in pFedKT or server in pFedHN).

## 5 THEORETICAL ANALYSIS AND PROOF

In this section, we analyze pFedKT's generalization and prove its convergence in theory.

### 5.1 ANALYSIS FOR GENERALIZATION

We refer to the theoretical analysis in Shamsian & et al (2021) and derive similar conclusions in Theorem 5.1, the detailed assumptions and derivations are illustrated in Appendix E.

**Theorem 5.1** *If one client has at least* $M = \mathcal{O}(\frac{1}{\epsilon^2}(emb_{dim} + HN_{size})log(\frac{rL_\omega L_\varphi}{\epsilon}) + \frac{1}{\epsilon^2}log(\frac{1}{\delta}))$ *samples, for hypernetwork* $\varphi$*, there is at least* $1 - \delta$ *probability that satisfies:* $|f(\varphi) - \hat{f}_{D_k}(\varphi)| \le \epsilon$.

where $f$ is loss function, $emb_{dim}$ and $HN_{size}$ are the input embedding dimension and size (parameter capacity) of hypernetworks (the hypernetwork is a fully connected model with chunk id as input embedding), $L_\omega, L_\varphi$ are assumed Lipschitz constants, $r, \epsilon$ are constants defined in derivation. *This Theorem reflects that pFedKT's generalization is impacted by both the hypernetwork's input embedding dimension and size*, Lipschitz constants $L_\omega$ and $L_\varphi$ which have been verified to marginally affect the hypernetwork's utility in Shamsian & et al (2021). So, we experimentally verify how the hypernetwork's input embedding dimension and size affect pFedKT's generalization in Sec. 6.3.1.

### 5.2 PROOF FOR CONVERGENCE

**Insight**. Shamsian & et al (2021) explains that the mapping of hypernetworks to generated target models is essentially similar to the principle of PCA dimension reduction. That is, the hypernetwork can be viewed as the *main component (core information)* of the target model after reducing dimension. Therefore, target models generated by hypernetworks would have a similar convergence rate to pure target models, as shown in preliminaries (Fig. 2 (b) in Sec. 3).

**Proof**. We refer to the convergence proof in Li et al. (2020), and derive the following Theorem (detailed proof is presented in Appendix E):

**Theorem 5.2** *Assuming* $E[f(\omega_T)]$ *is the average loss in the $T$-th round, $f^*$ is the minimum loss of $\omega$ during $T$ rounds; $\kappa, \gamma, B, C, \mu, L$ are defined constants in Li et al. (2020); $\omega_0$ is the initial model, $\omega^*$ is the optimal model with minimum loss. Then we can get:* $\mathbb{E}[f(\omega_T)] - f^* \le \frac{2\kappa}{\gamma+T}(\frac{B+C}{\mu} + 2L \cdot L_\varphi \sigma_4{}^2) \sim \mathcal{O}(1/T)$*, where $L_\varphi$ and $\sigma_4$ are constants defined in our extra assumptions.*

From Theorem 5.2, we conclude that *pFedKT has the same convergence rate* $\mathcal{O}(1/T)$ *with FedAvg*.

## 6 EXPERIMENTS

We implement pFedKT and all baselines with PyTorch and simulate their FL processes on NVIDIA GeForce RTX 3090 GPUs with 24G memory. We evaluate pFedKT on two image classification datasets: CIFAR-10/100 [1] (Krizhevsky & et al, 2009) with manually Non-IID division and a large real-world Non-IID dataset: Stack Overflow [2]. The Codes will be made public after acceptance.

---

[1] https://www.cs.toronto.edu/ kriz/cifar.html

[2] https://www.tensorflow.org/federated/api_docs/python/tff/simulation/datasets/stackoverflow/load_data

## 6.1 SETTINGS

**Datasets and Models**. Referring to the Non-IID data divisions in Shamsian & et al (2021); Charles & et al. (2021), we manually divide the three datasets into *Non-IID* distributions. Specifically, for CIFAR-10, we assign only 2 classes of data to each client, 50 clients totally; for CIFAR-100, we assign only 10 classes of data to each client, 50 clients totally; for Stack Overflow, we assign only posts from one author to each client, 100 clients totally. In CIFAR-10/100, one class allocated to different clients has different numbers and features. Specifically, if one class requires to be allocated into 10 clients, then we use $random.uniform(low, high)$ function ($low, high \in (0, 1)$) to produce 10 ratios of data counts and then generate this class's data subnets with different numbers and features to different clients. In Stack Overflow, each user's posts show naturally diverse in class and features. After dividing Non-IID data into each client, each client's local data is further divided into training set, evaluating set, and testing set with the ratio of 8:1:1, i.e., the testing set is stored locally in each client and shows a consistent distribution with local training set. We train a small CNN model and a large CNN model on the CIFAR-10 and 100 datasets in two image classification tasks, respectively, and train an LSTM model with the same structure as McMahan & et al (2017) on Stack Overflow dataset in a next-word prediction task. We use the same hypernetworks in three tasks. The structures of two CNN models and hypernetworks are shown in Appendix F.1, Tab. 2.

**Baselines**. We compare pFedKT with the following algorithms: 1) **Local Training**, in which each client trains its model locally. 2) **FedAvg**, a typical FL algorithm. 3) Cluster-based PFL methods: **HYPERCLUSTER**. 4) PFL methods with model mixup: **FedRep**, **FedPer**, **LG-FEDAVG**, **MAPPER**, **PartialFed**. 5) PFL methods with local aggregation: **FedFOMO**. 6) PFL methods with knowledge distillation: **FML**, **FedPHP**. 7) PFL methods related to our pFedKT: **MOON** with contrastive learning which regards GM as the anchor, **pFedHN** with the server's hypernetwork, and **Fed-RoD** with personalized headers generated by hypernetworks.

**Metrics**. We measure the trained private models' *mean* accuracy and denote it as *PM@Acc (%)*.

**Training Strategy**. We set grid-searched optimal FL hyperparameters for all algorithms: the client sampling rate $C$ is $0.1$; the learning rate of the local target model ($\eta_{NN}$) is $1e-2$, using the SGD optimizer with the momentum of $0.9$, weight decay of $5e-5$, and batch size of $64$, local epochs of $\{10, 50, 100\}$, and the hypernetwork's learning rate ($\eta_{HN}$) is $5e-3$; the total communication rounds are at most $500$. Our pFedKT's unique hyperparameters are reported in Appendix F.1, Tab. 3.

## 6.2 COMPARISONS WITH BASELINES

Limited to pages, here we only report the comparison results on the CIFAR-10/100 dataset, the results on the Stack Overflow dataset are recorded in Tab. 4 of Appendix F.2.1. Tab. 1 records PMs' mean accuracy of our pFedKT and all baselines, and Fig. 4 displays how the PMs' mean accuracy varies with rounds and 50 clients' individual PMs' accuracy after convergence.

**Results**. As shown in Tab. 1, *our pFedKT's mean PM accuracy outperforms all baselines*. On the CIFAR-10 dataset, pFedKT's mean PM's accuracy is $98.12\%$, which is $1.11\%$ improved than the second-highest PM@Acc $97.01\%$ achieved by pFedHN. On the CIFAR-100 dataset, pFedKT's mean PM's accuracy is $69.38\%$, increased $1.62\%$ than the second-highest PM@Acc $67.76\%$ achieved by FedPHP. The 100-classification task is a bit more complex than the 10-classification task, so it is inspiring that pFedKT fulfils a larger improvement of PM's accuracy on the CIFAR-100 dataset. Since pFedHN's global hypernetwork may not fully learn private models' knowledge in a more complex task while our pFedKT's local hypernetwork can well learn its own generated private model's knowledge and then still maintains the highest accuracy. Besides, in Fig. 4, it's obvious to see that: (1) our pFedKT converges to the highest PM's accuracy. (2) Overall, the 50 PMs' individual accuracies after convergence in pFedKT are better (lighter color) than the baselines, which demonstrates the highest personalization degree of pFedKT.

**Analysis**. The PM@Acc of pFedHN and Fed-RoD are second only to our pFedKT, which benefits from that: pFedHN uses the server's *large* hypernetwork to learn clients' private knowledge and transfer it to each client, promoting PM's feature extraction ability; Fed-RoD utilizes local hypernetworks to learn personalized *header's* knowledge, improving PM's prediction ability. FedPHP marginally improves PM due to linear cumulative knowledge; MOON fails to improve PM may be due to choosing GM as the initial anchor. *Profiting from transferring historical private knowledge via local hypernetworks into PMs and transferring global knowledge via contrastive learning into PMs, PM's personalization and generalization in available classes are both enhanced in pFedKT, so it achieves the state-of-the-art personalized PM's accuracy.*

Table 1: The PMs' mean accuracy of pFedKT and compared baselines on CIFAR-10 (Non-IID: 2/10) and CIFAR-100 (Non-IID: 10/100) datasets. For fair comparisons, we record the best *PM@Acc (%)* during 500 rounds for all algorithms. Bold: highest, italic: second-highest.

| Dataset | CIFAR-10 | CIFAR-100 |
|---|---|---|
| Local Training | 95.11 | 59.78 |
| FedAvg (McMahan & et al, 2017) | 96.42 | 65.73 |
| HYPCLUSTER (Mansour & et al, 2020) | 95.75 | 63.30 |
| FedRep (Collins & et al, 2021) | 96.74 | 66.64 |
| FedPer (Arivazhagan & et al, 2019) | 96.61 | 66.28 |
| LG-FEDAVG (Liang & et al, 2020) | 92.60 | 58.51 |
| MAPPER (Mansour et al., 2020) | 95.68 | 61.91 |
| PartialFed (Sun & et al, 2021) | 95.48 | 61.62 |
| FedFOMO (Zhang & et al, 2021) | 94.19 | 53.4 |
| FML (Shen et al., 2020) | 83.18 | 47.82 |
| FedPHP (Li & et al., 2021) | 96.60 | *67.76* |
| MOON (Li et al., 2021) | 89.21 | 48.64 |
| pFedHN (Shamsian & et al, 2021) | *97.01* | 67.41 |
| Fed-RoD (Chen & Chao, 2022) | 96.61 | 67.50 |
| **pFedKT (Ours)** | **98.12** | **69.38** |

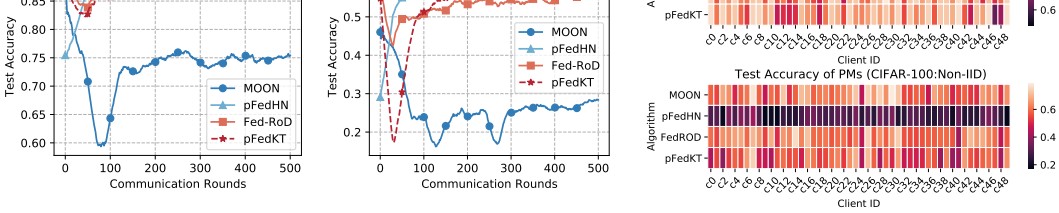

Figure 4: On CIFAR-10/100 datasets, left-two: the smoothed PMs' mean accuracy varies with communication rounds; right-two: 50 clients' individual PM's accuracy after convergence.

**Non-IID and Client Participating Rate**. In addition, we also verified that pFedKT presents superiority in highly Non-IID degrees and shows robustness to diverse client participation rates. The detailed experimental settings, results and analysis are reported in Appendix F.2.2.

## 6.3 CASE STUDY

In this section, we study the affects of several cases on pFedKT, which include: the hypernetwork's input embedding dimension and size, the weight of contrastive loss, the margin of triplet loss, and diverse loss functions. All case studies are executed within 500 rounds on CIFAR-10 (Non-IID: 2/10) and CIFAR-100 (Non-IID: 10/100) datasets. For stable comparisons, we record the average of *PM@Acc (%)* within the last 30 rounds.

### 6.3.1 INPUT EMBEDDING DIMENSION AND SIZE OF HYPERNETWORK

**A. Input embedding dimension**. Sec. 5.1 analyzed that the hypernetwork's input embedding dimension may affect pFedKT's generalization, so we experimentally explore its actual influence. Like Shamsian & et al (2021) computing the embedding dimension by $\lfloor 1 + N/\beta \rfloor$ ($N$ is the total number of clients, $N = 50$), we select $\beta \in \{1, 2, 3, 4, 10\}$, i.e., $emb_{dim} \in \{51, 26, 17, 13, 6\}$. Fig. 7 of Appendix F.3 displays that the accuracy of PM varies with different embedding dimensions, and the detailed values are recorded in Tab. 6 of Appendix F.3. From Fig. 7, we can see that: on the CIFAR-10 dataset, the embedding dimension has random effects on pFedKT; on the CIFAR-100 dataset, the PM's accuracy rises as the embedding dimension increases. However, the larger input embedding dimension has higher computation complexity, which slows down local training. In Appendix F.3, Tab. 6, when embedding dimensions are 13 and 51 on CIFAR-10/100 datasets respectively, pFedKT obtains the best PM.

**B. Size**. Sec. 5.1 also mentioned that pFedKT's generalization may be affected by the hypernetwork's size, so we vary the number of hypernetworks' *hidden layers* $\in \{1, 2, 3, 4, 5\}$, and results are reported in Fig. 7 and Tab. 7 of Appendix F.3. From Fig. 7, we find that the accuracy of PM drops as hypernetwork's size increases, which is probably due to that larger hypernetworks require more parameters to be trained, degrading their generalization. In Tab. 7, pFedKT performs the best PM when the number of the hypernetwork's hidden layers is 1, 2 on CIFAR-10, 100 datasets.

### 6.3.2 HYPERPARAMETERS IN CONTRASTIVE LEARNING

Next, we explore how the following key parameters of contrastive learning affect pFedKT: (1) $\mu$, which controls the weight of contrastive loss; (2) $\alpha$, the margin of triplet loss; and (3) different combinations of loss functions and distance measurements.

**A. Weight of contrastive loss.** We vary $\mu \in \{0, 0.0001, 0.001, 0.01, 0.1, .., 0.9\}$, and the results are reported in Fig. 7 and Tab. 8 of Appendix F.3. Tab. 8 shows that pFedKT with $\mu = 0.001, 0.0001$ obtain the best PM on CIFAR-10/100. Fig. 7 presents that when $\mu > 0.1$, PM's accuracy degrades obviously as $\mu$ rises, which may because that: a larger weight of contrastive loss and a smaller weight of supervised loss lead to PM's insufficient training, since PM accesses less supervised information from labels. *Hence, the weight of contrastive loss should be set smaller than that of supervised loss.*

**B. Margin of triplet loss.** We vary $\alpha \in \{0, 0.1, 1, 5, 10, 20, 30, 40, 50\}$, and the results are reported in Fig. 7 and Tab. 9 of Appendix F.3. Tab. 9 shows that pFedKT with $\alpha = 0.1, 5$ achieve optimal PM on CIFAR-10, 100 datasets, respectively. Fig. 7 displays that: *when $\alpha \geq 30$, PM's accuracy drops obviously* since larger $\alpha$ leads initialized PM to be overly biased to the immature GM, directly compromising PM's personalization. *Therefore, setting an appropriate $\alpha$ is necessary to balance the private knowledge and global knowledge transferred to PM.*

**C. Loss functions.** We also explore how diverse combinations of distance measurements and loss functions affect pFedKT, and the detailed experimental settings, results and analysis are given in Appendix F.3. From the results in Appendix F.3, Tab. 10, we conclude that *the combination of triplet loss and KL divergence we designed in pFedKT performs better model accuracy than others.*

### 6.4 ABLATION STUDY

pFedKT involves two important parts: (1) transferring private knowledge via hypernetwork (HN) and (2) transferring global knowledge via contrastive learning. To verify the effectiveness of each part, we conduct ablation experiments. We explore the following four cases: (A) Without part-(1,2), then pFedKT degenerates into FedAvg. (B) Only executing part-(1), i,e., only repeating the local training steps: generating PM by HN $\rightarrow$ training PM $\rightarrow$ updating HN's parameters. C) Only executing part-(2), viewing the PM initialized with GM as the anchor, GM as the positive item, and the last round's PM as the negative item, then this case degrades to MOON, but we still use triplet loss with KL divergence to compute contrastive loss. D) Executing both the two parts, i.e., pFedKT.

As shown in Appendix F.3, Tab. 11, case-(A) (i.e., FedAvg) has the lowest PM accuracy. Compared case-(B) with case-(A), the PM's accuracy is obviously improved, indicating that using hypernetworks to transfer private knowledge is reasonable. Compared case-(C) with case-(A), there are marginal improvements in PM's accuracy, reflecting that MOON takes limited improvements on model accuracy. Compared case-(D) with case-(B), pFedKT with contrastive loss takes slight accuracy improvement than without it. Fig. 8 in Appendix showed that pFedKT with contrastive loss has more stable convergence than w/o contrastive loss on CIFAR-10, and it converges faster to the highest accuracy within 300 rounds while pFedKT w/o contrastive loss requires 500 rounds for convergence on CIFAR-100. The above results indicate that the global knowledge transferred by contrastive loss enhances PM's generalization in available classes with lower computation cost. Case-(D) (i.e., pFedKT) achieved the highest PM's accuracy, demonstrating that the two-type knowledge transfers are necessary to enhance PM's personalization.

**Summary**. Overall, pFedKT fulfils the state-of-the-art personalized PM's performances. Besides, both the hypernetwork's input embedding dimension and size influence pFedKT's generalization. And it's necessary to select the proper weight of contrastive loss, the margin of triplet loss, and the combination of the contrastive loss function and distance measurement. Finally, ablation experiments verified the feasibility and effectiveness of both two types of knowledge transfer in pFedKT.

## 7 CONCLUDING REMARKS

In this paper, we proposed a novel personalized FL framework named pFedKT to boost personalized PMs. It consists of two types of knowledge transfer: a) transferring historical private knowledge to PMs by local hypernetworks, and b) transferring global knowledge to PMs through contrastive learning. The two-type knowledge transfer enables PMs to acquire both historical private knowledge and the latest global knowledge, which promotes PM's personalization and generalization in available classes simultaneously. Besides, we theoretically and experimentally verified that pFedKT's generalization is related to hypernetworks' input embedding dimension and size and also proved its convergence. Extensive experiments demonstrated that pFedKT achieves the state-of-the-art personalized PM's accuracy. In practice, pFedKT can be broadly applied to cross-silo FL scenarios.

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

# A    DEFINITION OF FL

**FedAvg** (McMahan & et al, 2017) is a typical federated learning algorithm. Assuming that there are $N$ clients in total, as shown in Fig. 1, the server samples fraction $C$ of clients $S^t$ ($|S^t| = N \cdot C = K$) to participate in FL and broadcasts the global model to them. The $k$-th client initializes its local model $\omega_k$ with the received global model and then trains it on local datasets $D_k$, the training objective is:

$$min\ F_k(\omega_k) = \frac{1}{n_k} \sum_{i \in D_k} f_i(\omega_k), \tag{5}$$

where $n_k = |D_k|$; $f_i(\omega_k) = \ell(\boldsymbol{x_i}, y_i; \omega_k)$, i.e., the loss of $i$-th instance $(\boldsymbol{x_i}, y_i)$ on the local model $\omega_k$. The local epoch is $E$, batch size is $B$, so local training executes $E\frac{n_k}{B}$ iterations. Then, clients upload trained local models to the server, and the server aggregates received local models to update the global model:

$$\omega = \sum_{k=0}^{K-1} \frac{n_k}{n} \omega_k, \tag{6}$$

where $n$ is the total number of instances owned by all clients. All the steps iteratively execute until the global model converges or the maximum round reaches.

# B    HOW TO USE HYPERNETWORKS TO GENERATE PARAMETERS FOR TARGET MODELS?

Taking the hypernetwork and CNN (target model) for CIFAR-10 in Tab. 2 as examples, here we introduce how we use the hypernetwork to generate parameters for target models. Since the volume $(2.e + 09)$ of CNN's parameters are more than the parameters amounts $(400)$ of the hypernetwork's output layer, so we first divide CNN's parameters into multiple chunks with ordered ids, each chunk has no more than $400$ ( hypernetwork's output dimension) parameters. Then we sequentially input the embedding of chunk id into the hypernetwork to get the corresponding chunk's target parameters. Finally, we concatenate the parameters from all chunks and reshape them with the CNN's parameter shape. To sum up, we call the hyperparameters multiple times to generate parameters for the target model in a stacked form. Since each client's local hypernetwork is unrelated to others, we applied the above parameter generation way for all clients in our pFedKT.

# C    HOW TO UPDATE HYPERNETWORK?

Assuming that the hypernetwork is $\varphi$, and it generates parameters for the target model $\omega = h(v; \varphi)$ ($v$ is the hypernetwork's input embedding). The hypernetwork $\varphi$ and the generated target model $\omega$ are trained in an *end-to-end* manner. Specifically, the generated target model $\omega$ first executes gradient descent, then the hypernetwork $\varphi$ is updated also through gradient descent. We assume that the loss function of the generated target model $\omega$ is $\ell(\omega) = \ell(h(v; \varphi))$, so we refer to Shamsian & et al (2021) and also utilize *chain rule* to derive the following equation:

$$\nabla_\varphi \ell(\omega) = \nabla_\omega \ell(\omega) \cdot \nabla_\varphi \omega = (\nabla_\varphi \omega)^T \cdot \nabla_\omega \ell(\omega). \tag{7}$$

For $\nabla_\omega \ell(\omega)$, we can use its first-order gradient approximation (Shamsian & et al, 2021; Zhang & et al, 2021) to represent as:

$$\nabla_\omega \ell(\omega) := \Delta \omega = \widehat{\omega} - \omega, \tag{8}$$

where $\widehat{\omega}$ and $\omega$ are the target models after/before training. So we replace $\nabla_\omega \ell(\omega)$ of Eq. 7 with Eq. 8, and get:

$$\nabla_\varphi \ell(\omega) = (\nabla_\varphi \omega)^T \cdot \Delta \omega \tag{9}$$

After computing the gradients of the hypernetwork $\varphi$, its parameters are updated through gradient descent, i.e.,

$$\varphi \leftarrow \varphi - \eta_{HN}(\nabla_\varphi \omega)^T \cdot \Delta \omega, \tag{10}$$

where $\eta_{HN}$ is the learning rate of the hypernetwork $\varphi$.

---

**Algorithm 1:** pFedKT

---

**Input:** $N$, number of clients; $K$, number of selected clients; $R$, total number of rounds; $\eta_{NN}$, learning rate of private target networks; $\eta_{HN}$, learning rate of private hypernetworks; $E$, local epochs, $B$, batch size; $\mu$, weight of contrastive loss, $\alpha$, margin in triplet loss

Randomly initialize the global model $\omega^0$, private model $\omega_k^0 = \omega^0$, private hypernetwork $\varphi_k^0$

**for** *each round t=0,1,..,R-1* **do**

    $S^t \leftarrow$ randomly select $K$ clients from $N$ clients

    **Clients execute:**

    **for** *each client $k \in S^t$* **do**

        Receive the latest global model $\omega^t$ form the server

        Utilize hypernetwork $\varphi_k^t$ to generate initial private model $\widetilde{\theta}_k^t$

        Train $\widetilde{\theta}_k^t$ on local dataset $D_k$ to get trained private model $\omega_k^t$ by contrastive learning:

        $\mathcal{B} \leftarrow$ split local dataset $D_k$ into batches of size $B$

        $\omega_k^t \leftarrow \widetilde{\theta}_k^t$

        **for** *each local epoch e from 1 to E* **do**

            **for** *each batch $b \in \mathcal{B}$* **do**

                Compute logits: $\mathcal{L}_{\omega^t} \leftarrow \mathcal{R}_{\omega^t}(b),\ \mathcal{L}_{\omega_k^t} \leftarrow \mathcal{R}_{\omega_k^t}(b), \mathcal{L}_{\omega_k^{t-1}} \leftarrow \mathcal{R}_{\omega_k^{t-1}}(b)$

                Compute distances: $d_{KL}^+ = d_{KL}(\mathcal{L}_{\omega^t} \parallel \mathcal{L}_{\omega_k^t}), d_{KL}^- = d_{KL}(\mathcal{L}_{\omega_k^{t-1}} \parallel \mathcal{L}_{\omega_k^t})$

                Compute contrastive loss: $\ell_{con} = \max\{d_{KL}^+ - d_{KL}^- + \alpha, 0\}$ // triplet loss

                Compute supervised loss: $\ell_{sup} = CrossEntropy$(output of $\omega_k^t(b)$, label)

                Complete complete loss function: $f = \mu * \ell_{con} + (1-\mu) * \ell_{sup}$

                Gradient descent: $\omega_k^t \leftarrow \omega_k^t - \eta_{NN}\nabla f$

            **end**

        **end**

        Store trained model $\omega_k^t$ locally

        Use trained private model $\omega_k^t$ and initial private model $\widetilde{\theta}_k^t$ to update hypernetwork:

        $\varphi_k^{t+1} \leftarrow \varphi_k^t - \eta_{HN}(\nabla_{\varphi_k^t}\theta_k^{t0})^T \Delta(\theta_k^{t0}, \omega_k^t)$ // according to Eq. 1

        Upload private trained model $\omega_k^t$ to the server

    **end**

    **Server executes:**

    Receive private models $[\omega_k^t, ...]$ from clients and aggregate them by:

    $\omega^{t+1} = \sum_{k=1}^K \frac{n_k}{n}\omega_k^t$ // $n_k$, number of $k$-th client's samples; $n$, total number of all clients' samples

    Send the updated global model $\omega^{t+1}$ to clients selected in the next round

**end**

**Return** personalized private models $[\omega_0^{R-1}, \omega_1^{R-1}, ..., \omega_{N-1}^{R-1}]$

---

## D  PFEDKT ALGORITHM

Here, we illustrate the detailed algorithm of pFedKT in Alg. 1.

## E  DETAILED THEORETICAL ANALYSIS AND PROOF

In this section, we detailed the assumptions and derivations for Theorem 5.1 and Theorem 5.2, respectively.

### E.1  DETAILED DERIVATIONS FOR THEOREM 5.1

We assume that the $k$-th client's local dataset $D_k = \{(\mathbf{x_i^{(k)}}, y_i^{(k)})\}_{i=1}^{|D_k|}, D_k \sim P_k$. Since we assume that the private model $\omega_k$ is nonlinear, the following derivations are adaptive to non-convex situations. Then the empirical loss and expected loss of the $k$-th client's private model $\omega_k$ can be denoted as:

$$\textbf{Empirical loss}: \hat{f}_{D_k}(\omega_k) = \frac{1}{|D_k|}\sum_{i=1}^{|D_k|}\ell(\mathbf{x_i^{(k)}}, y_i^{(k)}; \omega_k), \tag{11}$$

$$\textbf{Expected loss}: f(\omega_k) = \mathbb{E}_{P_k}[\ell(\mathbf{x^{(k)}}, y^{(k)}; \omega_k)]. \tag{12}$$

Since we utilize local hypernetworks to generate parameters for private models, $\omega_k = h(v; \varphi_k)$. So we can replace $\omega_k$ with $h(v; \varphi_k)$ in the above two losses, i.e.,

$$\textbf{Empirical loss}: \hat{f}_{D_k}(\varphi_k) = \frac{1}{|D_k|} \sum_{i=1}^{|D_k|} \ell(\mathbf{x_i^{(k)}}, y_i^{(k)}; h(v; \varphi_k)), \tag{13}$$

$$\textbf{Expected loss}: f(\varphi_k) = \mathbb{E}_{P_k}[\ell(\mathbf{x^{(k)}}, y^{(k)}; h(v; \varphi_k))]. \tag{14}$$

Since each client holds its own hypernetwork which is unrelated to other's hypernetworks, i.e., the hypernetwork's input embedding $v$ is independent of clients, so the variant is only $\varphi_k$

We refer to the assumptions about the parameters $\varphi$ of hypernetworks in Shamsian & et al (2021), in which the hypernetwork's parameters are bounded in a spherical space with radius $r$. Besides, we also assume the following two Lipschitz conditions:

**Assumption E.1** *The supervised loss $\ell_{sup}$ of the private model $\omega$ is Lipschitz smooth, i.e., $\ell_{sup}$ satisfies:*

$$\|\ell_{sup}(\boldsymbol{x}, y; \omega_1) - \ell_{sup}(\boldsymbol{x}, y; \omega_2)\| \le L_\omega \|\omega_1 - \omega_2\|. \tag{15}$$

As analyzed in Qian & et al. (2019); Chen et al. (2021); Pang et al. (2022), we can also assume that the triplet loss $\ell_{con}$ of pFedKT is Lipschitz smooth. So we can update the above assumption as:

**Assumption E.2** *The complete loss $\ell$ of the private model $\omega$ satisfies:*

$$\|\ell(\boldsymbol{x}, y; \omega_1) - \ell(\boldsymbol{x}, y; \omega_2)\| \le L_\omega \|\omega_1 - \omega_2\|. \tag{16}$$

**Assumption E.3** *The mapping from the hypernetwork $\varphi$ to the target private model $\omega$ is Lipschitz smooth, i.e.,*

$$\|h(v; \varphi_1) - h(v; \varphi_2)\| \le L_\varphi \|\varphi_1 - \varphi_2\|. \tag{17}$$

Among the above two Lipschitz conditions, $L_\omega, L_\varphi$ are Lipschitz constants.

Based on the above assumptions and the derived threshold of local data volume $M = \mathcal{O}(\frac{1}{\epsilon^2} log(\frac{\mathcal{C}(\epsilon, \mathbb{H}_l)}{\delta}))$ in Shamsian & et al (2021); Baxter (2000), $\mathcal{C}(\epsilon, \mathbb{H}_l)$ is the *covering number* of $\mathbb{H}_l$, $\mathbb{H}_l$ is parameterized by the hypernetwork $\varphi$. And the distance between two hypernetworks $\varphi_1, \varphi_2$ on dataset with distribution $P$ can be computed by:

$$\begin{aligned} \mathrm{d}(\varphi_1, \varphi_2) &= \mathbb{E}_{(\boldsymbol{x_i}, y_i) \sim P}[|\ell(h(v; \varphi_1)(\boldsymbol{x_i}), y_i) - \ell(h(v; \varphi_2)(\boldsymbol{x_i}), y_i)|] \\ &\le L_\omega \|\ell(h(v; \varphi_1)) - \ell(h(v; \varphi_2))\| \\ &\le L_\omega L_\varphi \|\varphi_1 - \varphi_2\|. \end{aligned} \tag{18}$$

Then we choose a $\epsilon$-covering parameter space, in which we can always find at least one neighbor $\varphi_2$ with $\frac{\epsilon}{L_\omega L_\varphi}$ distance to $\varphi_1$, so we can get:

$$\log(\mathcal{C}(\epsilon, \mathbb{H}_l)) = \mathcal{O}((emb_{dim} + HN_{size}) \log(\frac{r L_\omega L_\varphi}{\epsilon})), \tag{19}$$

where $emb_{dim}$ and $HN_{size}$ are the input embedding and parameter capacity of the hypernetwork.

Similar to Theorem 1 in Shamsian & et al (2021), we can conclude that our pFedKT's generalization is affected by the hypernetwork's input embedding and size, and also the above Lipschitz constants, as illustrated in Theorem 5.1.

### E.2 DETAILED DERIVATIONS FOR THEOREM 5.2

Based on above Lipschitz conditions, we further make the following assumptions:

**Assumption E.4** *The gradients and parameters of models are bounded, i.e.,*

$$\begin{aligned} \mathbb{E}[\|g_{(\omega)}\|^2] \le {\sigma_1}^2, \mathbb{E}[\|\omega\|^2] \le {\sigma_2}^2, \\ \mathbb{E}[\|g_{(\varphi)}\|^2] \le {\sigma_3}^2, \mathbb{E}[\|\varphi\|^2] \le {\sigma_4}^2, \end{aligned} \tag{20}$$

where $\varphi$ is the hypernetwork and $\omega$ is the target model with parameters generated by the hypernetwork $\varphi$, $\sigma_{1,2,3,4}$ are constants.

Li et al. (2020) have proved that FedAvg can converge to $\mathcal{O}(1/T)$ on Non-IID dataset when partial clients participate in FL, i.e.,

$$\mathbb{E}[f(\omega_T)] - f^* \leq \frac{2\kappa}{\gamma + T}\left(\frac{B + C}{\mu} + 2L\|\omega_0 - \omega^*\|^2\right), \tag{21}$$

where $\mathbb{E}[f(\omega_T)]$ is the average loss in the $T$-th round; $f^*$ is the minimum loss of $\omega$ during $T$ rounds' optimization; $\kappa, \gamma, B, C, \mu, L$ are constants assumed or derived in Li et al. (2020); $\omega_0$ is the initialized model, $\omega^*$ is the optima with $f^*$. Since the above conclusion has been proved in Li et al. (2020), here we no further detail it.

In our pFedKT, each client's private model $\omega$ is generated by its private hypernetwork $\varphi$, so we have:

$$\omega_0 = h(v; \varphi_0), \omega^* = h(v; \varphi^*). \tag{22}$$

Hence, Eq. (21) can be replaced as:

$$\mathbb{E}[f(\omega_T)] - f^* \leq \frac{2\kappa}{\gamma + T}\left(\frac{B + C}{\mu} + 2L\|h(v; \varphi_0) - h(v; \varphi^*)\|^2\right). \tag{23}$$

From Assumption E.3, the above equation can be further derived as:

$$\mathbb{E}[f(\omega_T)] - f^* \leq \frac{2\kappa}{\gamma + T}\left(\frac{B + C}{\mu} + 2L \cdot L_\varphi\|\varphi_0 - \varphi^*\|^2\right). \tag{24}$$

According to the Lipschitz condition in Assumption E.4, we can further get:

$$\mathbb{E}[f(\omega_T)] - f^* \leq \frac{2\kappa}{\gamma + T}\left(\frac{B + C}{\mu} + 2L \cdot L_\varphi\sigma_4^2\right) \sim \mathcal{O}(1/T), \tag{25}$$

where $L_\varphi$ and $\sigma_4$ are the constants defined in Assumption E.3 and Assumption E.4. Therefore, our pFedKT has the same convergence rate $\mathcal{O}(1/T)$ with FedAvg.

## F    EXPERIMENTAL DETAILS

### F.1    MODEL STRUCTURES AND HYPERPARAMETERS

We describe the structures of CNN models used on CIFAR-10/100 datasets and hypernetworks used for all tasks in Tab. 2. And we also report the detailed pFedKT's hyperparameters used in three tasks in Tab. 3.

Table 2: Structures of CNN models on CIFAR-10/100 datasets and hypernetworks used for CIFAR-10 and Stack Overflow datasets. Note: the hypernetwork used on CIFAR-100 dataset has 2 hidden layers, i.e., its structure: fc1 (emb_dim, 100) $\rightarrow$ fc2 (100, 100) $\rightarrow$ fc3 (100, 100), $\rightarrow$ fc4 (100, 400).

| | CNN (CIFAR-10) | | | CNN (CIFAR-100) | | | Hypernetwork | | |
| layer name | input size | output size | filter | input size | output size | filter | input size | output size | filter |
| --- | --- | --- | --- | --- | --- | --- | --- | --- | --- |
| conv1 | 3×32×32 | 16×28×28 | 5×5×16 | 3×32×32 | 16×28×28 | 5×5×16 | - | - | - |
| max pool1 | 16×28×28 | 16×14×14 | 2×2 | 16×28×28 | 16×14×14 | 2×2 | - | - | - |
| conv2 | 16×14×14 | 32×10×10 | 5×5×32 | 16×14×14 | 32×10×10 | 5×5×32 | - | - | - |
| max pool2 | 32×10×10 | 32×5×5 | 2×2 | 32×10×10 | 32×5×5 | 2×2 | - | - | - |
| fc1 | 800 | 1920 | - | 800 | 1920 | - | Emb_dim | 100 | - |
| fc2 | 1920 | 80 | - | 1920 | 80 | - | 100 | 100 | - |
| fc3 | 80 | 10 | - | 80 | 100 | - | 100 | 400 | - |

### F.2    MORE EXPERIMENTAL RESULTS OF COMPARISONS WITH BASELINES

Here, we also detailedly report the experimental results on Stack Overflow dataset and the comparison results of our pFedKT and four related advanced baselines on CIFAR-10/100 dataset with different Non-IID degrees and diverse client participation rates.

Table 3: The settings of pFedKT's hyperparameters in three tasks. emb_dim is the hypernetwork's input embedding dimension; n_hidden is the hypernetwork's number of hidden layers; $\mu$ is the weight of contrastive loss; $\alpha$ is the margin in triplet loss. Note that all the hyperparameters are *approximate optimal* due to our coarse-grained searching.

| Dataset | emb_dim | n_hidden | $\mu$ | $\alpha$ | contrastive loss | distance |
|---|---|---|---|---|---|---|
| CIFAR-10 (Non-IID) | | 1 | | 0.1 | | |
| CIFAR-100 (Non-IID) | 13 | 2 | 0.001 | 5 | triplet loss | KL divergence |
| Stack Overflow | | 1 | | 0.1 | | |

Table 4: The experimental results of our pFedKT and baselines on Stack Overflow dataset (a large natural real-world Non-IID dataset).

| Model@Acc | Local Training | FedAvg | FedPHP | MOON | pFedHN | Fed-RoD | pFedKT (ours) |
|---|---|---|---|---|---|---|---|
| PM@Acc | 15.71±2.30 | 23.52±0.05 | 23.64±0.03 | 23.79±1.41 | 25.19±0.01 | 24.98±0.15 | **25.33±0.01** |

### F.2.1 EXPERIMENTAL RESULTS ON STACK OVERFLOW DATASET

We report the experimental results on a large natural real-world Non-IID dataset *Stack Overflow* in Tab. 4. We can observe that our pFedKT presents the best PM's accuracy, again verifying its utility.

### F.2.2 COMPARISONS WITH BASELINES ON DATASETS WITH DIFFERENT NON-IID DEGREES

To explore how the Non-IID degree affects the performances of our pFedKT and four related advanced baselines, we allocate $\{1, 2, ..., 10\}$ classes of data into one client for the CIFAR-10 dataset and divide $\{10, 20, ..., 100\}$ classes of data into one client for CIFAR-100 dataset, and the results are reported in Fig. 5 and Tab. 5.

**Results**. From Fig. 5, we observe that PM's accuracy drops as the IID degree rises (the number of classes owned by one client rises). Since clients hold more classes of data, their PMs have lower preferences to each class (i.e., compromised personalization), so PMs' accuracy degrades. This result is also consistent with the argument in Shen et al. (2020): GM targets to improve its generalization but PM aims to improve its personalization, so Non-IID is beneficial to PM but harmful to GM. Our pFedKT presents superiority on *severely Non-IID* data, but performs marginally worse than baselines in high IID data, such as CIFAR-10 with class = $\{8, 9, 10\}$ and CIFAR-100 with class = $\{70, 80, 90, 100\}$. *Nevertheless, decentralized data held by devices participating in FL are often highly Non-IID (Kairouz & et al., 2021), hence our pFedKT's utility is still practical.*

**Analysis**. pFedHN uses a server's hypernetwork to learn local models' parameter distributions instead of simple weighted aggregation in FedAvg, when local datasets are more IID, the server's hypernetwork learn local knowledge more evenly and hence showing higher accuracy as IID degrees increase. FedPHP allows each client's linearly accumulated old local model to teach the received global model in a knowledge distillation paradigm, it actually trains the global model, so it adapts more to IID data. Our pFedKT performs worse than baselines in IID data because it emphasizes the local training of personalized private models and increasing IID degrees may compromise personalization.

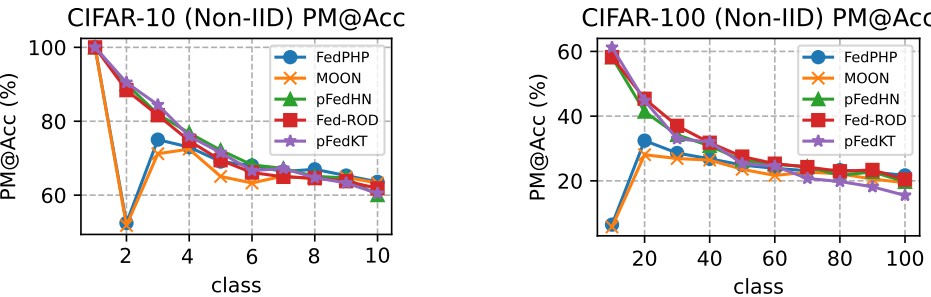

Figure 5: Test accuracy of PM on CIFAR-10/100 datasets varies with different Non-IID degrees.

Table 5: Numerical test accuracy of PM on CIFAR-10/100 datasets with different Non-IID degrees.

| | PM@Acc, CIFAR-10 (Non-IID) | | | | | | | | | |
|---|---|---|---|---|---|---|---|---|---|---|
| class | 1 | 2 | 3 | 4 | 5 | 6 | 7 | 8 | 9 | 10 |
| FedPHP | **100.00** | 52.46 | 75.02 | 72.99 | 69.09 | 68.01 | 66.40 | **66.99** | **65.30** | **63.60** |
| MOON | **100.00** | 51.78 | 71.22 | 72.46 | 65.06 | 63.29 | 65.18 | 64.36 | 64.84 | 63.27 |
| pFedHN | **100.00** | 90.03 | 81.93 | **76.82** | **72.17** | **68.20** | **67.23** | 65.10 | 64.65 | 60.02 |
| Fed-RoD | **100.00** | 88.41 | 81.63 | 74.71 | 69.45 | 66.12 | 64.95 | 64.61 | 63.78 | 61.84 |
| pFedKT | **100.00** | **90.48** | **84.44** | 75.94 | 71.51 | 66.64 | 67.21 | 64.76 | 63.29 | 60.47 |
| | PM@Acc, CIFAR-100 (Non-IID) | | | | | | | | | |
| class | 10 | 20 | 30 | 40 | 50 | 60 | 70 | 80 | 90 | 100 |
| FedPHP | 6.47 | 32.43 | 28.60 | 26.81 | 24.90 | 23.86 | 23.13 | **23.31** | 22.74 | **21.68** |
| MOON | 5.78 | 28.07 | 26.90 | 26.42 | 23.57 | 21.67 | 22.71 | 22.17 | 20.57 | 19.43 |
| pFedHN | 58.20 | 41.40 | 34.22 | 30.81 | 26.36 | 25.08 | **24.42** | 21.68 | 22.79 | 19.70 |
| Fed-RoD | 58.25 | **45.37** | **37.01** | 31.78 | **27.50** | **25.31** | 24.26 | 22.96 | **23.37** | 20.41 |
| pFedKT | **61.24** | 44.79 | 33.00 | **32.02** | 25.35 | 24.57 | 20.74 | 19.80 | 18.16 | 15.51 |

### F.2.3 COMPARISONS WITH BASELINES UNDER DIFFERENT CLIENT PARTICIPATION RATES

To evaluate the effects of our pFedKT and baselines under different client participation rates (i.e., fraction $C$), we conduct experiments on CIFAR-10/100 datasets with 50 clients. We vary fraction $C \in \{0.1, 0.2, ..., 1\}$ and report the results in Fig. 6. We can observe that: a) our pFedKT presents the highest PM accuracy in any fraction setting; b) pFedKT's PM accuracy is less affected by fraction. In addition, we also test our pFedKT and state-of-the-art Fed-RoD under the FL settings with 500 clients and *lower* $frac = 0.01$ client participation rate. The PM accuracy of Fed-RoD and our pFedKT are $79.79\%, 79.98\%$ on CIFAR-10 dataset and $30.85\%, 31.63\%$ on CIFAR-100 dataset. All the above results verify that pFedKT is robust to client participation rates.

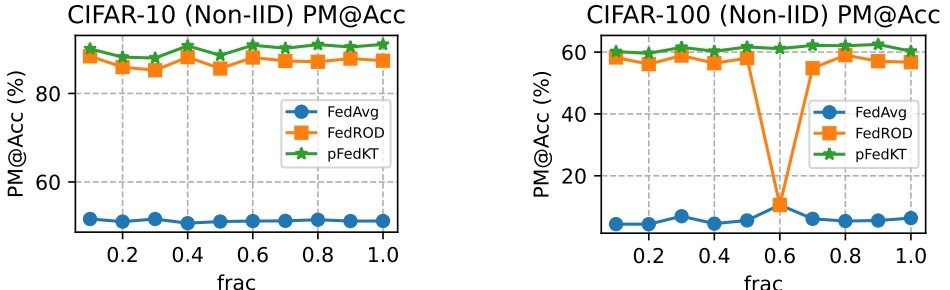

Figure 6: Test accuracy of PM varies with different client participation rates (fraction $C$) on CIFAR-10/100 datasets.

### F.3 DETAILED EXPERIMENTAL RESULTS IN CASE STUDY

Here we report the detailed experimental results of five cases on CIFAR-10 (Non-IID: 2/10) and CIFAR-100 (Non-IID: 10/100) datasets in Fig. 7 and Tab. 6-9.

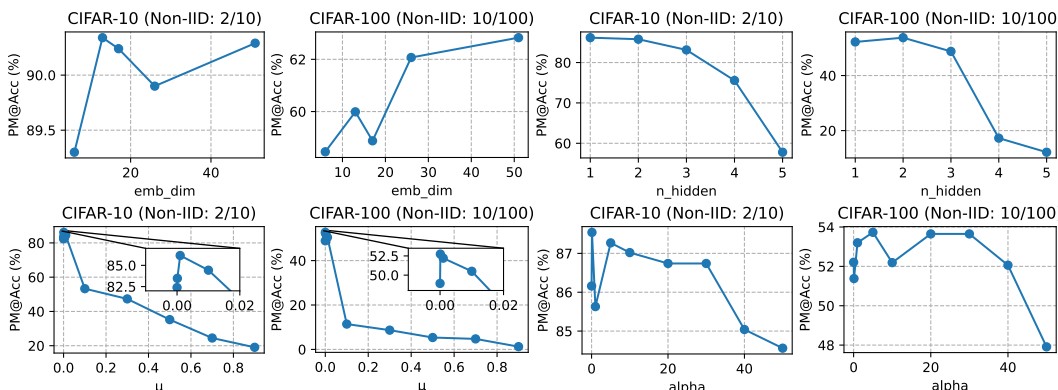

Figure 7: PMs' mean test accuracy varies with the hypernetwork's input embedding dimension and number of hidden layers (size), the weight $\mu$ of contrastive loss, and the margin $\alpha$ in triplet loss.

Table 6: The test accuracy of PM varies with *the hypernetwork's input embedding dimension.*

| Dataset | CIFAR-10 (Non-IID) | CIFAR-100 (Non-IID) |
|---|---|---|
| emb_dim | PM@Acc | PM@Acc |
| 6 | 89.30 | 58.46 |
| 13 | **90.34** | 59.99 |
| 17 | 90.24 | 58.88 |
| 26 | 89.90 | 62.07 |
| 51 | 90.29 | **62.83** |

Table 7: The test accuracy of PM varies with *the hypernetwork's number of hidden layers (size).*

| Dataset | CIFAR-10 (Non-IID) | CIFAR-100 (Non-IID) |
|---|---|---|
| n_hidden | PM@Acc | PM@Acc |
| 1 | **86.16** | 52.21 |
| 2 | 85.78 | **53.76** |
| 3 | 83.13 | 48.77 |
| 4 | 75.56 | 17.26 |
| 5 | 57.79 | 12.15 |

Table 8: The test accuracy of PM varies with *the weight $\mu$ of contrastive loss.*

| Dataset | CIFAR-10 (Non-IID) | CIFAR-100 (Non-IID) |
|---|---|---|
| $\mu$ | PM@Acc | PM@Acc |
| 0 | 82.39 | 48.92 |
| 0.0001 | 83.47 | **52.74** |
| 0.001 | **86.16** | 52.21 |
| 0.01 | 84.43 | 50.50 |
| 0.1 | 53.30 | 11.41 |
| 0.3 | 47.32 | 8.67 |
| 0.5 | 35.23 | 5.35 |
| 0.7 | 24.49 | 4.73 |
| 0.9 | 19.07 | 1.24 |

Table 9: The test accuracy of PM varies with *the margin $\alpha$ in triplet loss.*

| Dataset | CIFAR-10 (Non-IID) | CIFAR-100 (Non-IID) |
|---|---|---|
| alpha | PM@Acc | PM@Acc |
| 0 | 86.16 | 52.21 |
| 0.1 | **87.54** | 51.38 |
| 1 | 85.63 | 53.20 |
| 5 | 87.27 | **53.73** |
| 10 | 87.02 | 52.19 |
| 20 | 86.74 | 53.65 |
| 30 | 86.74 | 53.65 |
| 40 | 85.04 | 52.06 |
| 50 | 84.56 | 47.91 |

## C. Loss functions in contrastive learning

There are various ways to measure the distances between two models' logits vectors, such as KL divergence, $1-$ (cosine similarity), L2 norm, and MSE (the square of L2 norm). There are also diverse contrastive loss functions, such as triplet loss (Schroff & et al., 2015), loss in MOON (Li et al., 2021), and etc. In fact, we pursue PM to be close to GM via contrastive learning, so a naive way is to add the euclidean distance (L2 norm) between the GM and PM as the regularization to the supervised loss, like FedProx (Li & et al, 2020). We evaluated several combinations of the above distance measurements and loss functions, and the results are reported in Tab. 10.

We can observe that the combination of triplet loss and KL divergence that we designed in pFedKT achieves the best PM on the CIFAR-10 dataset. pFedKT with loss used in MOON (Li et al., 2021) gets the highest PM's accuracy on the CIFAR-100 dataset but lower PM's accuracy on CIFAR-10 dataset than pFedKT with triplet loss and KL divergence. Other combinations also show worse model performances than our designed loss for pFedKT. It's worth noting that pFedKT with contrastive loss has higher accuracy than with L2 regularization, which verifies using contrastive loss to keep new PM (generated by HN) away from old PM is necessary and does prevent being into local optima.

Table 10: PM's test accuracy varies with *combinations of distance measurements and loss functions.*

| Loss Function | Dataset / Distance Measurement | CIFAR-10 PM@Acc | CIFAR-100 PM@Acc |
|---|---|---|---|
| Triplet Loss | KL divergence | **86.16** | 52.21 |
| | $1-$ (cosine similarity) | 84.38 | 53.48 |
| | L2 norm | 85.94 | 52.41 |
| | MSE | 59.18 | 12.96 |
| Moon loss | cosine similarity | 84.66 | **53.81** |
| L2 Regularization | KL divergence | 84.84 | 52.96 |
| | cosine similarity | 84.39 | 53.22 |
| | L2 norm | 85.99 | 51.34 |
| | MSE | 83.55 | 49.17 |

Table 11: Results of ablation experiments. "HN→PM" denotes private knowledge transfer by hypernetworks, "CL" represents global knowledge transfer through contrastive learning.

| Case | Dataset / HN→PM | CL | CIFAR-10 PM@Acc | CIFAR-100 PM@Acc |
|---|---|---|---|---|
| A | ✗ | ✗ | 51.64±0.02 | 4.59±0.39 |
| B | ✓ | ✗ | 90.10±0.51 | 61.23±0.67 |
| C | ✗ | ✓ | 51.78±0.07 | 16.40±0.03 |
| D | ✓ | ✓ | **90.34±0.12** | **61.66±0.08** |

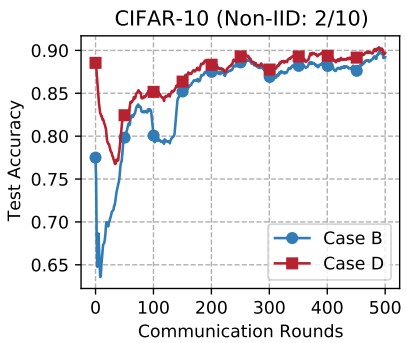 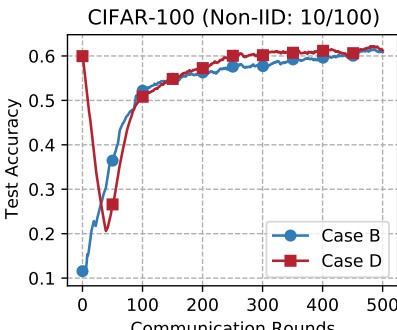

Figure 8: PMs' mean test accuracy of case B (HN→PM w/o CL) and case D (HN→PM w/ CL) vary with communication rounds on CIFAR-10/100 datasets.

## G PFEDKT'S COMPUTATIONAL COMPLEXITY, STORAGE COST AND MODEL PERFORMANCES

In this section, we compare pFedKT, state-of-the-art pFedHN and other baselines in computational complexity, storage overhead, and model performances.

### G.1 COMPUTATIONAL COMPLEXITY

Taking the CNN (CIFAR-10) and the hypernetwork in Tab. 2 as examples, we denote CNN (CIFAR-10) as *NN* (i.e., target private model) and the hypernetwork as *HN(small)*. In pFedHN (Shamsian & et al, 2021), one large hypernetwork is deployed on the server, which we denotes as *HN(large)*. For fair comparisons, we set the same number of hidden layers of HN(small) and HN(large), i.e., the structure of HN(large) is: fc1(emb_dim=13, 100) → fc2(100,100) → fc3(100, NN_Param). We compute the tree models' parameter capacity and computational overhead for one-time forward operation, and the results are recorded in Tab. 12

Table 12: The three models' parameter capacity and computational overhead (FLOPs) for one-time forward operation.

| Metric | Parameter Capacity | | | Computational Overhead (FLOPs) | | |
|---|---|---|---|---|---|---|
| Layers | NN | HN (small) | HN (large) | NN | HN (small) | HN (large) |
| conv1 | (5*5*3+1)*16 | - | - | ((3*5*5)+(3*5*5-1)+1)*16*28*28 | - | - |
| conv2 | (5*5*16+1)*32 | - | - | ((16*5*5)+(16*5*5-1)+1)*32*10*10 | - | - |
| fc1 | (800+1)*1920 | (13+1)*100 | (13+1)*100 | 2*800*1920 | 2*13*100 | 2*13*100 |
| fc2 | (1920+1)*80 | (100+1)*100 | (100+1)*100 | 2*1920*80 | 2*100*100 | 2*100*100 |
| fc3 | (80+1)*10 | (100+1)*400 | (100+1)* 1706458 | 2*80*10 | 2*100*400 | 2*100*1706458 |
| Total (number) | 1706458 | 51900 | 172363758 | 7822400 | 102600 | 341314200 |
| Total (MB/GB) | 6.5096 MB | 0.1980 MB | 657.5156 MB | 0.0291 GB | 0.0004 GB | 1.2715 GB |

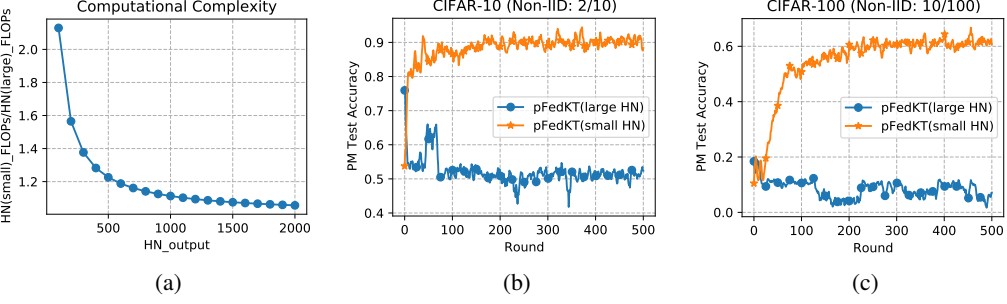

(a)      (b)      (c)

Figure 9: (a): the ratio of computational complexity (FLOPs) between HN (small) and HN (large) varies with the output dimension of HN (small); (b-c): PM's accuracy of pFedKT with HN (small) and HN (large) on CIFAR-10/100 datasets varies with rounds.

From Tab. 12, it requires to call HN (small) (NN_Paras/output of HN (small)) *times* to generate the whole parameters for one NN in a stacking form, so generating NN by HN (small) requires (NN_Paras/output of HN(small))*HN (small)_FLOPs, i.e., $(1706458/400) * 102600 = 1.6306$ GB

FLOPs. Using HN (large) to generate one NN *once* consumes 1.2715 GB FLOPs, as shown in Tab. 12. Hence the ratio of the former and the latter is about $1.28\times$, and it tends to be $1\times$ as the output dimension of HN (small) increases, as shown in Fig. 9 (a). *In short, our pFedKT consumes comparable computational cost to pFedHN.*

Besides, pFedHN updates the server's HN (large) once if it receives one private model. Using one private model to update HN (large) and then using the updated HN (large) to generate parameters for the private model consume 1.2715*2 GB FLOPs. When multiple private models reach the server simultaneously, *computational blocking* may occur due to the high computational complexity of HN (large). Whereas, our pFedKT deploys one HN (small) on each client. *From the perspective of computational complexity, our pFedKT inherently offloads the training tasks with the server's large hypernetwork in pFedHN to clients' sub-tasks, which tackles the above blocking issue.*

## G.2 STORAGE OVERHEAD

pFedHN's server requires about 657 MB to store the HN (large), while our pFedKT's $N$ clients consume $N * 0.1980$ MB storage cost. When the number $N$ of clients participating in FL is about 3321, our pFedKT has a comparable storage cost to pFedHN. But in the cross-silo FL scenario, there are often a few companies or institutions joining in FL (Kairouz & et al., 2021), so *our pFedKT has obvious strength than pFedHN in terms of storage cost.*

## G.3 MODEL PERFORMANCE

We have compared the model performances of pFedHN and our pFedKT in Sec. 6.2, here we do not repeat it. As illustrated above, we require to call HN (large) *once* or HN (small) *multiple times* to generate parameters for one NN. Here, we also test our pFedKT with private HN (small) and private HN (large) on CIFAR-10/100 datasets, and the results are shown in Tab. 13 and Fig. 9 (b)-(d). It can be seen that pFedKT with HN (large) shows similar model performances with FedAvg, which is consistent with our conclusion of the case study on HN's size in Sec. **??**: larger HN is harder to train, hence showing worse model accuracy. *This evaluation also verifies the strength of our calling HN (small) multiple times on model performances.*

Table 13: The results of pFedKT with HN (large) and HN (small) on CIFAR-10/100 datasets.

| Dataset | CIFAR-10 | CIFAR-100 |
|---|---|---|
| Method | PM@Acc | PM@Acc |
| FedAvg | 51.64 | 4.59 |
| pFedHN | 90.03 | 58.2 |
| pFedKT (small HN) | **90.34** | **61.66** |
| pFedKT (large HN) | 51.67 | 5.04 |

## G.4 COMPARED WITH MORE BASELINES

We also compare our pFedKT with the typical FedAvg and related MOON, Fed-ROD in computational overhead and storage efficiency.

**pFedKT v.s. FedAvg**. Compared with FedAvg, pFedKT introduces extra local computation of a local hypernetwork and contrastive loss, and additionally stores a local hypernetwork and a previous local model. But, as show in Tab. 1 pFedKT's accuracy is improved 38.7% in CIFAR-10 and 57.07% in CIFAR-100. Therefore, in cross-silo FL scenario, increased cost compared with obvious accuracy improvement is acceptable for participating enterprises and institutions.

**pFedKT v.s. MOON**. Compared with MOON, pFedKT introduces the computation of a local hypernetwork and additionally store a local hypernetwork for each client. But pFedKT improves 38.56% and 56.41% accuracy than MOON in CIFAR-10 and CIFAR-100, respectively.

**pFedKT v.s. Fed-ROD**. Compared with FedAvg, Fed-ROD's extra computations involve that using a local hypernetwork to generate parameters for a personalized header and locally train personalized branch, while it also requires to store a local hypernetwork additionally. Compared with Fed-ROD, our pFedKT introduces the following computations: using a local hypernetwork to generate parameters for a complete local target model and calculate contrastive loss. Since pFedKT's local

hypernetwork generates parameters for a whole local model and Fed-ROD's local hypernetwork generates parameters for the personalized header of a local model, pFedKT incurs relatively higher computation cost. In addition, pFedKT requires to store a local hypernetwork and a previous local model, but Fed-ROD only stores a local hypernetwork, so pFedKT has a slightly higher storage overhead. However, similar to the above analysis, pFedKT improves $0.31\%$ and $3.46\%$ accuracy improvements than Fed-ROD in CIFAR-10/100 datasets, which is acceptable.

