# OpenReview forum: "pFedKT: Personalized Federated Learning via Knowledge Transfer"
_ICLR.cc/2023/Conference — Submitted to ICLR 2023_

### Official Review · Reviewer_5c7g · 2022-10-23

**Confidence:** 4
**Correctness:** 3
**Technical Novelty And Significance:** 2
**Empirical Novelty And Significance:** 3
**Recommendation:** 5

**Clarity, Quality, Novelty And Reproducibility:**

Clarity: The paper has a good presentation.

Quality: The experiments need further improvement. Please see Strength And Weaknesses.

Novelty: Fair. The usage of hypernetwork is not new [2, 3]. Applying contrastive loss in local training has also been exploited [4].

[2] Personalized Federated Learning using Hypernetworks. ICML 2021.

[3] ON BRIDGING GENERIC AND PERSONALIZED FEDERATED LEARNING FOR IMAGE CLASSIFICATION. ICLR 2022.

[4] Model-Contrastive Federated Learning. CVPR 2021.

Reproducibility: Code is not provided.



**Strength And Weaknesses:**

Strength: The paper works on an important problem of FL. The idea is clear and the baselines in the experiments are comprehensive.

Weaknesses:
1. The improvement of pFedKT is low compared with other personalized FL approaches such as pFedHN. From Table 5, the accuracy of pFedKT is even lower than pFedHN in many cases.

2. Each client has a fixed number of classes in the experiments, which is not practical in most applications. More real-world federated datasets can be used in experiments.

3. The local iterations per round is fixed to 100 in the experiments. Experiments with different numbers of local iterations are necessary since the baselines may not work well with a large number of local iterations.

4. The paper assumes that the contrastive loss is Lipschitz smooth and refers to [1] to support the assumption. I cannot see a clear relation between the assumption and the reference. The paper needs to provide more explanation about it.

[1]  Softtriple loss: Deep metric learning without triplet sampling.

5. The paper compares many baselines in the experiments. However, some of them are not personalized FL approaches (e.g., FedProx, SCAFFOLD). The authors may consider removing them to save space.


**Summary Of The Paper:**

The paper proposes pFedKT to address the Non-IID data issue in federated learning. pFedKT is based on personalized FL with knowledge transfer. During local training, it utilizes a hypernetwork to generate a local model. The local model is updated with a contrastive loss as regularization, which limits the distance between the local model and global model to incorporate the knowledge of global model into local model. The hypernetwork is then updated with the updates in local model such that the hypernetwork can absorb the knowledge of the existing local models to generate the local model for the next round. Experiments show that pFedKT achieves better accuracy than the other baselines.

**Summary Of The Review:**

The paper proposes a new PFL algorithm. While the presentation is good, I think the experiments need further improvement and the theoretical analysis needs more justification.

---

### Official Review · Reviewer_fgAG · 2022-10-23

**Confidence:** 5
**Correctness:** 3
**Technical Novelty And Significance:** 2
**Empirical Novelty And Significance:** 2
**Recommendation:** 5

**Clarity, Quality, Novelty And Reproducibility:**

The paper has some unclear statements. For example, it is unclear how test data is set up. It seems that they are local at clients.
Function $f$ is never defined. It appears to be the loss function. The papers states 'where f is loss function' but then why not using the notation from Section 3.3.

It appears that they deal with the horizontal setting but I don't think this is clearly stated.

**Strength And Weaknesses:**

Strengths:
The use of prior local models is novel.

Weaknesses:
The theoretical analysis is unclear and the connections with non-iid is unclear.
The computational experiments do not show extensive improvements and are flaky.

**Summary Of The Paper:**

The authors tackle the problem of non-iid data in FL. They propose the use of hypernetworks and the target network to cope with this. Clients use a contrastive loss to mitigate non-iid.
The key idea is for clients to utilize prior local weights and new global weights obtained from the server. They also provide a theoretical convergence analysis.
The computational experiments show that the proposed algorithm outperforms the benchmarks.

**Summary Of The Review:**

The use of prior local models is definitely a nice idea however I think the authors don't do much with it. The application of the contrastive loss is not new and thus not a contribution.

A have issues with theoretical results.
Theorem 4.1 require a large number of samples which pretty much goes against non-iid. Clearly a large number of samples does not imply idd but it does exclude many non-iid situations. Theorem 4.1 is loosely stated but it seems to show convergence to global optimal in absence of convexity. This clearly cannot hold.
The same remark holds for Theorem 4.2.

The experiments are limited. The data is non-iid only in terms of class distribution but not in terms of features. This is clearly quite limited.

Table 1: Improvements over local training are 'too good to be true.' I guess it questions if local training has been optimized. For CIFAR-10 the improvement definitely doesn't impress (90.34 vs 90.03).

There is one definitely relevant reference that is missing: Y. Xue, D. Klabjan, and Y. Luo. Aggregation Delayed Federated Learning.

---

### Official Review · Reviewer_Luyb · 2022-10-31

**Confidence:** 3
**Correctness:** 2
**Technical Novelty And Significance:** 2
**Empirical Novelty And Significance:** 2
**Recommendation:** 3

**Clarity, Quality, Novelty And Reproducibility:**

### Clarity
* The main idea of the proposed global knowledge transfer scheme with the contrastive loss is not clear enough (See Weaknesses above for details).
* In regards to the efficiency, the advantage of the proposed models compared to others is not sufficiently clear (See Weaknesses above for details).
* In Figure 1 (b), it is unclear how to measure the global and local model accuracies. Specifically, in main experiments, the authors report the test accuracy based on the local private data with the local model; then, how to measure the global model accuracy, and how to compare this global model accuracy to the local model accuracies?

### Quality
* The experimental quality of the analyses in Section 5.3 might be low, since the authors conduct the analyses with the unconverged model.
* In Section 2.1, "clients train the received GM on local datasets from scratch" should be tone-downed, since the client trains the local model from the globally aggregated model, which indeed contains the information for the local model; not training from the scratch.

### Novelty
* The novelty is mild, since, for the knowledge transfer, the concepts of hypetnetworks and contrastive learning are already proposed in the previous work, such as pFedHN, Fed-ROD, and MOON; however, the differences are faithfully and sufficiently discussed in the related work section as well as other sections.

### Reproducibility
* The authors do not provide the source code that lowers the reproducibility of this paper; however, the authors plan to release the source code after the acceptance. Therefore, the reproducibility will be probably high.

**Strength And Weaknesses:**

### Strengths
* The proposed historical knowledge transfer scheme with the hypernetwork brings the performance improvement for the personalized federated learning tasks.
* The authors make effort to theoretically analyze the generalization and convergence bounds of the proposed pFedKT, while they are mostly inspired by the previous work [1] though.
* The authors perform extensive analyses on the proposed pFedKT, by varying the hyperparameters, and by ablating the knowledge transfer mechanisms.

### Weaknesses
* The proposed global knowledge transfer scheme is not convincing enough in terms of both the motivation- and experiment-sides.
* * At first, the authors argue that, by maximizing the similarity between the locally updated model and the globally aggregated model while minimizing the similarity between the locally updated model and the previously updated local model, the proposed pFedKT improves the generalization performance. However, this design choice is not convincing, since the authors already transfer the historical knowledge in the previously updated local model with the hypernetworks, meanwhile, the historical knowledge is negatively considered (i.e., historical information is avoided) in the contrastive learning loss. Therefore, two objectives are conflicts in the federated learning.
* * In the experiment-side, the proposed global knowledge transfer scheme also does not bring the meaningful performance improvement, i.e., not much helpful for the personalized federated learning. For example, in Figure 7, the proposed global knowledge transfer scheme based on the contrastive loss does not bring the performance improvement. Similarly, in Table 11, the results w/ and w/o contrastive losses are very similar.
* * Furthermore, in the experimental-side, it is unclear whether the proposed global knowledge transfer scheme can provide better generalization ability empirically. While the authors provide the theoretical result for the generalization bound, I suggest authors to include additional empirical results, if possible, which makes it more convincing.
* In Section 3.4, the authors only compare the computational and storage efficiencies of the proposed pFedKT against the complex pFedHN model. It is meaningful to discuss the efficiencies of the most basic FedAvg model, as well as the other contrastive- and hypernetwork-based federated learning models, such as MOON and Fed-ROD.
* Also, in Section 3.4, the authors argue that the proposed pFedKT has obvious strengths against the pFedHN model, since the pFedHN baseline has the larger hypernetwork in the server-side, while the proposed pFedKT has smaller hypernetworks in the client-side. However, this is not convincing, since if we have 1,000 clients, we have 1,000 individual hypernetworks distributed to 1,000 clients, and, in the global view, the size of 1,000 individual hypernetworks would be larger than the size of one hypernetwork in the server.
* The analysis results in Section 5.3 may be problematic. The authors argue the proposed pFedKT can converge during 100 rounds, therefore, conduct analyses with 100 rounds; however, pFedKT does not converge until 100 rounds, as shown in Figure 4.
* Since knowledge distillation-based federated learning methods share similar sprits to the proposed knowledge transfer-based federated learning model: the knowledge distillation allows the local/global models to transfer their knowledge effectively, I suggest authors to compare such knowledge distillation-based federated learning baselines: FedPHP, FML and KT-pFL, discussed in the related work section.

---

[1] Personalized federated learning using hypernetworks, ICML 2021.

**Summary Of The Paper:**

This paper aims to improve the performance of personalized federated learning, and, for which, the authors propose two knowledge transfer schemes. In particular, the historical knowledge learned in the local clients is transferred from the hypernetwork, which stores the knowledge of previous local models, to the current local model. Also, the global knowledge, obtained by the aggregation of local models, is transferred to the local clients based on the contrastive learning loss, where the similarity between the updated local model and the aggregated global model is maximized while the similarity between the updated local model and the previous local model is minimized. The authors validate the performance of their model, named as pFedKT, on two image (i.e., CIFAR-10/100) and one language (i.e., Stack Overflow) datasets, on which the pFedKT outperforms relevant personalized federated learning baselines.

**Summary Of The Review:**

The main idea of the proposed global knowledge transfer scheme based on the contrastive loss is not convincing (See Weaknesses above), and there are some improvement points, such as efficiency analyses in Section 3.4 which are not sufficient, and model analyses which are perhaps conducted without the model convergence. Therefore, I cannot recommend the acceptance.

---

### Comment · Area_Chair_Pcwh · 2022-12-04
**The end of the discussion period approaching soon**

Dear Reviewers,

The end of the discussion period is quickly approaching. Could you please go over the final responses from the authors and the reviews from the others, and provide feedback to the authors?

Thanks,
AC

---

### Decision · Program_Chairs · 2023-01-20

**Decision:**

Reject

**Justification For Why Not Higher Score:**

The proposed method lacks novelty as it is incremental over pFedHN, and its effectiveness over pFedHN is also inconclusive from the current set of experimental results.

**Justification For Why Not Lower Score:**

None

**Metareview: Summary, Strengths And Weaknesses:**

The paper proposes a personalized federated learning framework with local hyper networks that generates personalized local model parameters, with inter-client knowledge transfer across the local personal models at different training rounds, as well as contrastive-loss based knowledge transfer between the global aggregated model and the personal models. The authors provide theoretical analyses of the method's generalization and convergence, and experimentally validate the proposed personalized FL framework on non-IID tasks generated out of three benchmark datasets. The experimental results show that the proposed method outperforms existing personalized FL algorithms, and the authors further experimental analyses in the appendix shows the effectiveness of the proposed knowledge transfer scheme.

In the initial reviews, all reviewers leaned toward rejection. The reviewers in general consider the overall knowledge transfer scheme as nice and intuitive, the experimental analyses as extensive, and the paper-well written. However, they are also concerned with the following weaknesses:

- The novelty of the proposed pFedKT framework over pFedHN [1] is limited, which also proposes a personalized FL framework based on hypernetwork, but the hypernetwork resides on the server instead of at each local clients.
- The contrastive loss-based knowledge transfer has been already explored in [2], which further weakens the novelty of the work.
- The method also obtains very small performance gains over the personalized FL baselines, especially over pFedHN, even on non-IID datasets it mainly targets.
- The theoretical analyses are based on restrictive assumptions and the experimental settings are artificial.

The authors provided responses to the reviewers, but the reviewers were not convinced that their concerns on the limited novelty and the effectiveness of the method has been addressed away and kept their original ratings. The authors may need to further develop their method or include discussion of how pFedKT differs from pFedHN, to address this critical concern on the novelty and the effectiveness of their method over pFedHN.

[1] Personalized Federated Learning using Hypernetworks, ICML 2021
[2] Model-Contrastive Federated Learning. CVPR 2021.